# Impacts of historical climate and land cover changes on fine particulate matter (PM$_{2.5}$) air quality in East Asia between 1980 and 2010

*Yu Fu[1,*], Amos P. K. Tai[2,*], Hong Liao[3]*

[1] *Climate Change Research Center (CCRC), Chinese Academy of Sciences, Beijing 100029, China*
[2] *Earth System Science Programme and Graduate Division of Earth and Atmospheric Sciences, Faulty of Science, Chinese University of Hong Kong, Hong Kong*
[3] *School of Environmental Science and Engineering, Nanjing University of Information Science & Technology, Nanjing, 210044, China*

*Corresponding authors: Yu Fu (fuyu@mail.iap.ac.cn), Amos P. K. Tai (amostai@cuhk.edu.hk)*

**Abstract.** To examine the effects of changes in climate, land cover and land use (LCLU), and anthropogenic emissions on fine particulate matter (PM$_{2.5}$) between the 5-year periods 1981-1985 and 2007-2011 in East Asia, we perform a series of simulations using a global chemical transport model (GEOS-Chem) driven by assimilated meteorological data and a suite of land cover and land use data. Our results indicate that climate change alone could lead to a decrease in wintertime PM$_{2.5}$ concentration by 4.0-12.0 µg m$^{-3}$ in northern China, but to an increase in summertime PM$_{2.5}$ by 6.0-8.0 µg m$^{-3}$ in those regions. These changes are attributable to the changing chemistry and transport of all PM$_{2.5}$ components driven by long-term trends in temperature, wind speed and mixing depth. The concentration of secondary organic aerosol (SOA) is simulated to increase by 0.2-0.8 µg m$^{-3}$ in both summer and winter in most regions of East Asia due to climate change alone, mostly reflecting higher biogenic volatile organic compound (VOC) emissions under warming. The impacts of LCLU change alone on PM$_{2.5}$ (-2.1 to +1.3 µg m$^{-3}$) are smaller than that of climate change, but among the various components the sensitivity of SOA and thus organic carbon to LCLU change (-0.4 to +1.2 µg m$^{-3}$) is quite significant especially in summer, which is driven mostly by changes in biogenic VOC emissions following cropland expansion and changing vegetation density. The combined impacts show that while the effect of climate change on PM$_{2.5}$ air quality is more pronounced, LCLU change could offset part of the climate effect in some regions but exacerbate it in others. As a result of both climate and LCLU changes combined, PM$_{2.5}$ levels are estimated to change by -12.0 to +12.0 µg m$^{-3}$ across East Asia between the two periods. Changes in anthropogenic emissions remain the largest contributor to deteriorating PM$_{2.5}$ air quality in East Asia during the study period, but climate and LCLU changes could lead to a substantial modification of PM$_{2.5}$ levels.

## 1 Introduction

Over the recent decades atmospheric particulate matter (PM, or aerosols) has received considerable attention in East Asian countries due to its impacts on regional air quality, human health and climate change. A number of projection studies have examined the effects of changes in anthropogenic emissions and climate on future PM air quality globally and in East Asia (Fiore et al., 2012; Jiang et al., 2013), but there still exist large uncertainties arising from the complex interactions between aerosol chemistry, meteorology and the underlying land cover, especially for East Asia which is expected to undergo tremendous land use change in the next few decades (Hurtt et al., 2011). A better understanding of how PM formation and removal have historically been shaped by meteorological and land surface conditions in East Asia would be useful to help to better project the future evolution of PM air quality. In this work, we use a chemical transport model driven by past meteorological and land surface data to evaluate the individual and combined effects of climate and land cover changes in East Asia over 1980-2010, and compare these effects with that of increasing anthropogenic emissions. This attribution of East Asia air quality trends in the past would shed light on their potential course of evolution in the coming few decades, and provide valuable information for policymaking concerning public health, land use and climate management.

Of particular public health concern is fine particulate matter ($PM_{2.5}$), defined to be suspended liquid or solid particles with a diameter of 2.5 μm or less. $PM_{2.5}$ has been shown to have detrimental effects on human health, leading to increased mortality related to cardiovascular diseases and lung cancer (Krewski et al., 2009; Silva et al., 2013; Fang et al., 2013). $PM_{2.5}$ is also associated with poor visibility and haze (Wang et al., 2014), and plays a significant role in modifying the Earth's energy budget (IPCC, 2013). $PM_{2.5}$ has a variety of sources that depend on the chemical components, which include sulfate, nitrate, ammonium, black carbon (BC), organic carbon (OC), sea salt, and mineral dust. Air quality degradation with elevated $PM_{2.5}$ in East Asia is primarily attributable to increasing anthropogenic emissions of their precursors. For instance, Wang et al. (2013) reported that annual mean sulfate-nitrate-ammonium concentrations increased by 60% in China from 2000 to 2006, which mainly resulted from the 60% and 80% increases in sulfur dioxide ($SO_2$) and nitrogen oxides ($NO_x$) emissions, respectively. Yang et al. (2015) found that the decadal trends of aerosol outflow from East Asia were dominated by the trends and variations in anthropogenic emissions, which could account for about 86% of the decadal trend in $PM_{2.5}$

outflow over 1986-2006.
Air pollution associated with $PM_{2.5}$ is also strongly sensitive to weather conditions and
therefore influenced by climate change (Jacob and Winner, 2009; Tai et al., 2010; Fiore et
al., 2012). Meteorological conditions affect the production, transport and deposition of $PM_{2.5}$
components and their precursors. The effects of climate change on $PM_{2.5}$ are complex due to
the widely varying sensitivities of its components to different meteorological factors. For
example, higher temperature can enhance sulfate concentration due to faster $SO_2$ oxidation
(Dawson et al., 2007; Jacob and Winner, 2009), while nitrate and OC concentrations
decrease because higher temperature shifts more of these semivolatile components from the
particle to gas phase (Liao et al., 2006; Kanakidou and Tsigaridis, 2007). This is further
complicated by the covariation of temperature with cold-frontal passages, which are an
important ventilating mechanism for air pollutants (Leibensperger et al., 2008; Tai et al.,
2012a; 2012b). In general, changes in ventilation (e.g., wind speed and direction, mixing
depth) significantly modify aerosol dispersion and transport. Zhu et al (2012) suggested that
the decadal-scale weakening of the East Asian summer monsoon could have increased
aerosol concentrations in eastern China mostly due to changes in circulation patterns.
Furthermore, higher humidity generally promotes the formation of ammonium nitrate
(Dawson et al. 2007; Tai et al., 2010), and all $PM_{2.5}$ components are very sensitive to
precipitation, which provides a major sink via scavenging (Dawson et al. 2007).
The tropospheric concentrations of $PM_{2.5}$ are also influenced by land cover and land use
(LCLU) changes. Vegetation represents an important source of biogenic volatile organic
compounds (VOC), especially isoprene and monoterpenes, which are major precursors of
secondary organic aerosols (SOA). SOA can be the major contributor to aerosols especially
in remote regions far away from industrial influence (Carslaw et al., 2010), but can also be
significant in many urban areas due to high year-round anthropogenic and summertime
biogenic VOC emissions. For instance, Ding et al. (2014) investigated the origins of SOA in
various Chinese regions based on observations from 14 sites during the summer of 2012, and
found that biogenic isoprene was the major contributor ($46\pm14\%$) to secondary OC in every
site. In addition, vegetation and land surface characteristics may further modulate
atmospheric aerosols by controlling soil $NO_x$ emissions and the dry deposition of both gases
and particles within the planetary boundary layer.
The impacts of climate and land cover changes on $PM_{2.5}$ on a multidecadal scale have
been quantified to various extents using chemical transport models (CTM) driven by
assimilated meteorological data or simulated meteorological fields from general circulation

models (GCM). Jeong and Park (2013) found that sulfate-nitrate-ammonium concentrations decreased by 4% in summer but increased by 7% in winter in eastern China over 1985-2006 as a result of meteorological changes alone. Jiang et al. (2013) showed that following climate change alone under the IPCC A1B scenario, different aerosol species over China would generally be altered by -1.5 to +0.8 μg m$^{-3}$, and PM$_{2.5}$ concentration is projected to change by 10-20% in eastern China. Tai et al. (2013) reported that in China, climate change together with climate- and CO$_2$-driven natural vegetation changes would change annual mean surface SOA by -0.4 to +0.1 μg m$^{-3}$ over 2000-2050 under the IPCC A1B scenario, while anthropogenic land use change can increase SOA by up to 0.4 μg m$^{-3}$ over the same period. Wu et al. (2012) also predicted that changes in natural vegetation and anthropogenic land use over 2000-2050 would lead to higher summertime SOA over East Asia. Most of these studies, except Jeong and Park (2013), are concerned with the effects of future climate change and/or land cover change on aerosols. Previous studies that focus on the effects of historical LCLU change on East Asian aerosols are few. Fu and Liao (2014) suggested that surface SOA might decrease by as much as 0.4 μg m$^{-3}$ (-20%) between the late 1980s and mid-2000s over China due to changes in biogenic emissions induced by LCLU (mainly) and climate (to a lesser extent) changes. However, the overall role of LCLU change in controlling regional PM$_{2.5}$ and its composition via biogenic emissions and deposition, especially under the simultaneous influence of climate change and CO$_2$ fertilization, is still poorly understood.

Because of climate change, rapid economic development and other human activities, LCLU in East Asia has undergone remarkable changes in the past 30 years. Especially in China, some major economic reforms and land use policies have been implemented since December 1978, which together with simultaneous changes in climate have resulted in a whirlwind of changes in the terrestrial environments of China. Based on satellite-derived images and surveys, LULC changes in China from late 1980s to the mid-2000s are characterized by an expansion of urban areas, deserts and bare lands, and an overall decrease in vegetation coverage (Fu and Liao, 2014). In this study, we use the GEOS-Chem global chemical transport model (CTM) driven by past land cover data and meteorological fields from a single and coherent set of assimilated meteorology (MERRA), which covers the period 1979-present. We quantify the impacts of historical changes in climate, land cover and land use on PM$_{2.5}$ air quality in East Asia between two 5-year periods: historical period, 1981-1985 (referred to as "1980"), and the present day, 2007-2011 (referred to as "2010"). We consider 5-year averages in each of these periods to account for interannual variability.

We also compare the effects of climate and LCLU changes with the contribution from
anthropogenic emissions over the same time frame. The findings would shed light on the
possible climate and land use "penalties" or benefits that might have exacerbated or offset
the effect of anthropogenic emissions in the past, and provide a constraint for future air
quality projections, which currently have large uncertainties for East Asia.

## 2 Methods and model description

We perform a series of model experiments to simulate aerosols using the GEOS-Chem
global CTM (version 9-02) with assimilated meteorology and integrated historical land cover
data. The modeling framework in this study is the same as that used in Fu and Tai (2015). In
brief, GEOS-Chem performs fully coupled simulations of ozone-$NO_x$-VOC (Bey et al.,
2001) and aerosol chemistry (Park et al., 2003, 2004; Pye et al., 2010). In this study, GEOS-
Chem is driven by the assimilated meteorological data from Modern Era Retrospective-
analysis for Research and Applications (MERRA) with a horizontal resolution of 2.0°
latitude by 2.5° longitude, and a vertical resolution of 47 levels. Aerosol species simulated
include sulfate, nitrate, ammonium, organic carbon, black carbon, sea salt, and mineral dust.
Inorganic aerosol thermodynamic equilibrium calculations are based on the ISORROPIA II
scheme of Fountoukis and Nenes (2007). SOA formation is based on the reversible gas-
aerosol partitioning of VOC oxidation products (Chung and Seinfeld, 2002, Liao et al.,
2007) with precursors including isoprene, monoterpenes, alcohols, and aromatic
hydrocarbons (Henze et al., 2008). The wet deposition scheme for water-soluble aerosol
species is described by Liu et al. (2001). Model details for other relevant modules and
emission inventories can be found in Fu and Tai (2015).
The land cover dependence of atmospheric chemistry is mainly encapsulated in two land
cover inputs, namely leaf area index (LAI) and land or plant functional types (as a single
categorical value or as fractional coverage in each grid cell), mostly through their effects on
biogenic VOC and soil $NO_x$ emissions, and dry deposition velocities. The emissions of
biogenic VOC species in each grid cell are determined by the canopy-scale emission factors
multiplied by various activity factors that account for variations in temperature, light, leaf
age and LAI, using online the Model of Emissions of Gases and Aerosols from Nature
(MEGAN) module (Guenther et al., 2012) embedded in GEOS-Chem with the emissions
time-step set to 30 minutes. Soil $NO_x$ emission follows the parameterization of Yienger and
Levy (1995) and Hudman et al. (2012), which includes a physical representation of key soil
processes derived from field measurements, with dependence on vegetation types,
temperature, precipitation history, fertilizer use, and a canopy reduction factor. The reservoir
of nitrogen associated with manure and chemical fertilizer remains unchanged between 1980
and 2010 by using the fixed inventory for fertilizer and manure emissions from Potter et al.
(2010). Dry deposition follows the resistance-in series scheme of Wesely (1989) as
implemented by Wang et al. (1998), and is dependent on species properties, land cover types
and meteorological conditions. It uses the Olson land cover classes with 76 land types
(Olson, 1992) reclassified into 11 land types. Aerosol dry deposition follows Zhang et al.
(2001) as described by Pye et al. (2009). The land cover inputs used for this study are
derived from a fusion of various datasets, including the Moderate Resolution Imaging
Spectroradiometer (MODIS) land cover product (MCD12Q1), the National Land Cover
Dataset (NLCD) for China, harmonized historical land use for Representative Concentration
Pathways (RCP) from Hurtt et al. (2011), and the global LAI product from Liu et al. (2012).
See Fu and Tai (2015) for detailed description of these land cover inputs.
We conduct a 5-year simulation in the present-day period (2007-2011) with the
corresponding meteorological variables, emissions, and land cover and land use as the
control simulation [*CTRL*]. Sensitivity simulations are conducted for: (1) [*S_CLIM*]:
historical (1981-1985) climate with present-day land cover inputs and emissions (scaled to
2005 levels) used in [*CTRL*]; (2) [*S_LCLU*]: historical land cover inputs with present-day
climate and emissions used in [*CTRL*]; (3) [*S_COMB*]: historical climate and land cover
inputs but with present-day emissions used in [*CTRL*]; and (4) [*S_ANTH*]: historical
emissions scaled to 1985 levels but with present-day climate and land cover inputs.
Additional sensitivity simulations perturbing certain meteorological variables while keeping
the rest at present-day levels are conducted to examine which meteorological factors have
been the most important for shaping different $PM_{2.5}$ components.

## 3 Simulated spatiotemporal variations of $PM_{2.5}$ concentrations

Simulated seasonal mean surface concentrations of sulfate ($SO_4^{2-}$), nitrate ($NO_3^-$),
ammonium ($NH_4^+$), black carbon (BC), organic carbon (OC), and total $PM_{2.5}$ (sum of sulfate,
nitrate, ammonium, BC and OC) in East Asia averaged over 2007-2011 from the [*CTRL*]
simulation are shown in Fig. 1. Simulated $PM_{2.5}$ is high over the eastern regions of East Asia
(east of 110 °E), especially in eastern and central China, where $PM_{2.5}$ is in the range of 60-90
$\mu g\ m^{-3}$ in all seasons. Nitrate (~33%), sulfate (~24%), ammonium (~19%) and OC (~22%)

are the major components of annual mean $PM_{2.5}$ in eastern China (Table S1). Based on the measurements from 16 sites in China, Zhang et al. (2012) reported that sulfate (~16%), OC (~15%), nitrate (~7%), ammonium (~5%) and mineral aerosol (~35%) are majorities of the total $PM_{10}$ concentration. The contribution of each species to total $PM_{2.5}$ is generally overestimated compared with nationwide measurements from Zhang et al. (2012), probably due to the exclusion of mineral dust and sea salt in this work, and the inadequacy of current emission inventories.

Among all seasons, simulated sulfate is the highest in summer (JJA) mainly due to enhanced photochemical oxidation of $SO_2$. The maximum summertime sulfate concentration of 25-35 $\mu g\ m^{-3}$ is found in the North China Plain, while the largest sulfate concentration in other seasons is within the range of 15-25 $\mu g\ m^{-3}$ over much of eastern and central China (Fig. 1). Nitrate shows a different seasonal variation with a maximum in winter (DJF) and minimum in summer, as the low wintertime temperature promotes ammonium nitrate formation. The maximum wintertime nitrate concentration is in the range of 30-40 $\mu g\ m^{-3}$ around central China (Fig. 1). The spatial distribution of seasonal ammonium concentration is similar to that of nitrate and sulfate, within a range of 5-20 $\mu g\ m^{-3}$ in all four seasons over the domain of study. The concentrations of OC and BC in the eastern regions of East Asia are the largest in winter, with maximum values of 20-25 $\mu g\ m^{-3}$ and 5-10 $\mu g\ m^{-3}$, respectively, reflecting higher anthropogenic emissions associated with domestic heating in winter. Our simulated aerosol concentrations and distributions show general agreement with previous studies in East Asia (Fu et al., 2012; Wang et al., 2013; Jeong and Park, 2013; Lou et al., 2014), demonstrating the ability of GEOS-Chem to capture the spatial variations of different $PM_{2.5}$ species despite biases in the absolute concentrations. The model biases of simulated annual mean sulfate, nitrate, ammonium, BC and OC in East Asia are -10%, +31%, +35%, -56% and -76%, respectively (Fu et al., 2012; Wang et al., 2013).

The distribution of simulated seasonal surface SOA concentration is shown in Fig. 2. Total SOA is the sum of SOA from the oxidation of biogenic VOC (including isoprene, monoterpenes, sesquiterpenes) and anthropogenic VOC (including benzene, toluene and xylene). In winter, simulated SOA is less than 0.5 $\mu g\ m^{-3}$ over the northern regions of East Asia, where biogenic emissions, temperature, and radiation are lower than in southern regions (south of 25 °N). In summer, simulated SOA over the eastern parts of East Asia are within the range of 2.0-7.0 $\mu g\ m^{-3}$ with the highest concentration in the lower and middle reaches of Yangtze River and east of Sichuan province. SOA distributions in spring (MAM) and fall (SON) are similar, within the range of 2.0-5.0 $\mu g\ m^{-3}$ over central and southeastern

China. Figure 2 also shows that SOA from biogenic emissions is the major contributor to total SOA in East Asia, larger than the anthropogenic contribution by nearly an order of magnitude in all seasons.

Of the total OC in China, about 6-27% is attributable to SOA (Table S1), which is lower than the fraction (~55-60%) found by Zhang et al. (2012). Fig. 3a compares the simulated and observed July-September mean SOA concentrations in 14 sites over China. The observed concentrations are taken from Ding et al. (2014), which are measured at 5 urban sites, 7 rural or sub-urban sites, and 2 remote sites around China in 2012. Simulated SOA is underestimated by about 60%. The simulated secondary organic carbon (SOC) to OC ratios are also compared with the observed ratios from Zhang et al. (2012) (Fig. 3b), which are measured at 14 sites in China during 2006-2007. The simulated seasonal mean SOC contributions range from 14% to 54% over China, with relatively higher contribution (54%) in summer but much lower (14%) in winter, while the observed ratios exhibit little seasonality between 51-57%. The simulated SOC/OC ratios agree generally well with the measurements in summer with a low-bias of 5%, while that in winter, spring, and autumn are underestimated by about 74%, 56%, and 41%, respectively (Fig. 3b). This suggests that SOA is underestimated in these seasons, likely reflecting uncertainties not only in the VOC emission inventories but also SOA formation mechanism. In this study, we consider SOA formation from absorptive partitioning of semivolatile organic compounds, but previous studies have suggested the potential importance of heterogeneous uptake of dicarbonyls (Fu et al., 2009) and oxidation of gas-phase semivolatile primary organic compounds and intermediate VOC (e.g., naphthalene) (Pye et al., 2010), which may constitute potentially large SOA sources but are not included in our study.

## 4 Impacts of climate change alone on $PM_{2.5}$

Climate change alone ([*CTRL*] − [*S_CLIM*]) can substantially alter the simulated concentrations of $PM_{2.5}$ and its components in East Asia between 1980 and 2010. Here we focus on the changes in winter and summer (Fig. 4 and Fig. 5, respectively). The most significant decrease of $PM_{2.5}$ occurs in winter (Fig. 4), with the maximum decrease in the North China Plain by up to 12.0 μg m$^{-3}$, while wintertime $PM_{2.5}$ over southeastern and central-western China (30°-40°N; 100°-110°E) increases by up to 4.0 μg m$^{-3}$, due to climate change alone. We find that the spatial pattern of changes in summertime $PM_{2.5}$ due to climate change alone is mostly reversed (Fig. 5), being enhanced by as much as 8.0 μg m$^{-3}$ in the

North China Plain and central-western China, but reduced by up to 8.0 μg m$^{-3}$ in central and
southern China.
Climate-driven changes in wintertime PM$_{2.5}$ are primarily driven by the changing
concentrations of sulfate, nitrate and ammonium induced by meteorological changes (Fig. 4).
Wintertime sulfate increases by up to 3.0 μg m$^{-3}$ over southeastern and central-western
China, but decreases by up to 2.0 μg m$^{-3}$ in the North China Plain. In southeastern and
central-western China, the simulated increase in wintertime sulfate is mostly a result of
regionally reduced surface wind speed and planetary boundary layer (PBL) (which reduce
ventilation and mixing), but also in part due to increased temperature (which accelerates SO$_2$
oxidation). In other regions, especially within the North China Plain, the simulated decrease
in sulfate reaches a maximum of -2.0 μg m$^{-3}$, likely reflecting increased surface wind speed.
Simulated nitrate and ammonium concentrations decrease in most of eastern parts of China,
which can be explained by the elevated temperature, decreased RH, enhanced wind speed
and PBL between the two periods (Fig. 4). The spatial patterns of changes and our sensitivity
simulations suggest that increased temperature has contributed the most broadly to the
reduction in ammonium nitrate formation, but depending on region, changes in wind speed
and PBL might have either substantially enhanced (most of northeastern, northern and
central China) or partly counteracted (e.g., southeastern China) the reduction in ammonium
nitrate, with RH playing only a minor role. OC and BC changes generally follow the same
patterns as those for sulfate, reflecting influence from the same suite of meteorological
variables.
Climate-driven changes in summertime total PM$_{2.5}$ are also dominated by changes in the
inorganic components, which generally show an opposite sign of changes to that in winter.
Summertime sulfate, nitrate and ammonium increase over northern China, the North China
Plain, and part of northwestern and eastern China, but decrease elsewhere (Fig. 5). In much
of central, southern and northeastern China, decreased sulfate, nitrate and ammonium are
attributable to the significantly increased PBL, which enhances mixing and dilution despite
reduced wind speed. Increased temperature also partly contributes to lower ammonium
nitrate. In northern and northwestern China, however, the large increase in sulfate is likely
driven by regionally reduced PBL and wind speed. The significantly enhanced nitrate and
ammonium concentrations in much of northern and eastern China are shaped by less
ventilation driven by wind speed in the North China Plain, further modulated by regional
cooling and increased RH around Shandong province (Fig. 5). We also find that simulated
summertime OC and BC increase in much of eastern China, reflecting a combination of
increased temperature and reduced wind speed, except in south-central China where OC is
reduced likely by enhanced PBL mixing.
Climate change alone could lead to increased SOA concentration in both winter and
summer in East Asia by as much as +0.8 μg m$^{-3}$ (Fig. 6). The climate-driven changes in SOA
are primarily due to changes in temperature that influence biogenic VOC emissions (Liao et
al., 2006). In winter, with enhanced isoprene and monoterpene emissions due to warming,
SOA concentration in the southern parts of China increases by up to 0.4 μg m$^{-3}$. In summer,
the simulated SOA concentration changes within the range of -0.5 to +0.8 μg m$^{-3}$, mostly
attributable to the biogenic emission changes but also partly modulated by transport changes,
similar to the pattern of OC changes (Fig. 5).

## 5 Impacts of land cover and land use change alone on PM$_{2.5}$


Figure 7 represents the impacts of 1980-2010 land cover and land use change alone on
PM$_{2.5}$ concentrations ([*CTRL*] − [*S_LCLU*]). We find that although the LCLU change effects
on all PM$_{2.5}$ components are generally smaller than the climate change effects, LCLU
change can in part modify (either exacerbate or offset) the sensitivity of PM$_{2.5}$ to climate
change. In most of the eastern parts of East Asia, wintertime PM$_{2.5}$ concentration increases
by up to 1.3 μg m$^{-3}$ as a result of LCLU change alone (Fig. 7a). In contrast, LCLU change
alone leads to a decrease in PM$_{2.5}$ by up to -2.1 μg m$^{-3}$ over much of China in summer
except in some of the southern parts (Fig. 7b). Such changes are mostly attributable to
changes in nitrate, ammonium and OC; BC is largely unaffected by LCLU change.
We find that with LCLU change alone, nitrate in winter increases by up to 0.6 μg m$^{-3}$
around central China (~30 °N). Such an increase is largely driven by reduced HNO$_3$ and NO$_2$
dry deposition following a decrease in wintertime LAI (Fig. S1 and Fig. S2 in supplementary
materials). The changes in ammonium follow the changes of nitrate, with which they are
chemically linked, and are partly due to reduced NH$_3$ dry deposition. In summer, the sign of
changes in nitrate and ammonium are mostly reversed. Nitrate decreases by as much as -1.2
μg m$^{-3}$ in the North China Plain, mostly driven by the enhanced HNO$_3$ and NO$_2$ dry
deposition resulting from enhanced summertime LAI (thus vegetation density) (Fig. S1 and
Fig. S3), overshadowing the effect of increased soil NO$_x$ emission from cropland expansion.
See Fu and Tai (2015) for more discussion on East Asian land cover change.
LCLU change effects on OC are relatively minor in winter (-0.1 to +0.2 μg m$^{-3}$), but are
significant in summer (-0.4 to +1.0 μg m$^{-3}$) since both LAI and plant functional type (PFT)

changes can significantly affect the emissions of biogenic VOCs, which are the major precursors to SOA especially in summer. In much of the North China Plain, central and northeastern China where deforestation and cropland expansion (in terms of PFT changes) have been the most rapid (Fig. S1), the effect of cropland expansion appears to dominate over that of enhanced summertime grid cell-averaged LAI in modifying biogenic emissions, leading to a decrease in OC that largely reflects a reduction in biogenic emissions (Fig. 8). The concentration of OC increases elsewhere, especially in southwestern and southern China where reforestation has been observed and increased summertime LAI further enhances the increase in biogenic emissions (Fig. S1). Figure 8 shows the contribution to surface SOA concentration from LCLU change alone, which largely follows the spatial pattern of OC changes and reflects the underlying changes in biogenic VOC emissions. Summertime SOA in summer increases by more than 1.0 $\mu g\ m^{-3}$ in southern and southwestern China between 1980 and 2010, but decreases by up to 0.4 $\mu g\ m^{-3}$ in other parts of China. In winter, LCLU change increases SOA by up to 0.4 $\mu g\ m^{-3}$ around Guizhou province, but leads to only negligible decreases in SOA in much of the rest of East Asia due to the small biogenic VOC emissions in winter.

# 6 Combined impacts of climate, land cover and land use changes vs. anthropogenic emissions

With anthropogenic emissions fixed at present-day levels, the changes in wintertime $PM_{2.5}$ resulting from both climate and LCLU changes combined between 1980 and 2010 are in the range of -12.0 to +6.0 $\mu g\ m^{-3}$ in East Asia, with the maximum decrease found in the North China Plain (Fig. 9a). In summer, changes in $PM_{2.5}$ are within the range of -8.0 to +12.0 $\mu g\ m^{-3}$ under the combined effects of climate and LCLU changes, with an enhancement of 4.0-12.0 $\mu g\ m^{-3}$ in the North China Plain. The changes of $PM_{2.5}$ and its components are largely driven by climate change. For SOA alone, the combined effects of climate change and LCLU changes modify summertime SOA by -0.6 to +1.2 $\mu g\ m^{-3}$, reflecting comparable contribution from both climate and LCLU changes, which can either exacerbate or offset each other depending on the region. For instance, in southwestern China, climate change alone might decrease SOA (Fig. 6), but LCLU change could more than offset the climate effect there (Fig. 9b).

Fig. 9c shows the effects on total $PM_{2.5}$ and SOA of changes in anthropogenic emissions alone ([*CTRL*] - [*S_ANTH*]), which we find as expected to be the dominant factor shaping

PM$_{2.5}$ air quality in East Asia over 1980-2010. In both summer and winter, PM$_{2.5}$ is
simulated to increase on average by 37% and 54% in East Asia, respectively, resulting from
changes in anthropogenic emissions. From 1985 to 2005, anthropogenic emissions of NO$_x$,
CO, SO$_2$, NH$_3$, OC, and BC have increased by 180%, 143%, 52%, 50%, 36% and 46%,
respectively, over East Asia (Table S2). Such emission-driven changes in PM$_{2.5}$ would be
partially offset in winter but substantially enhanced in summer by climate- and LCLU-driven
changes in the most polluted regions (e.g., in the vicinity of the North China Plain) between
1980 and 2010.

## 7 Conclusions and discussion

We simulate the effects of changes in climate, land cover and land use (LCLU), and
anthropogenic emissions between the two 5-year periods 1981-1985 (historical) and 2007-
2011 (present-day) on the surface concentrations of total PM$_{2.5}$ and its components including
sulfate, nitrate, ammonium, organic carbon (OC), and black carbon (BC) in East Asia using
the GEOS-Chem chemical transport model driven by assimilated meteorological data and a
suite of satellite- and survey-derived LCLU data. GEOS-Chem is shown to capture the
spatial and seasonal variations of different PM$_{2.5}$ species in the present day despite some
significant biases in the absolute concentrations. The present-day secondary organic aerosol
(SOA) concentration is underestimated in comparison with measurements in China. The
volatile organic compound (VOC) emission inventory and SOA formation mechanism might
represent the major sources of uncertainty for SOA simulation in the model.
With anthropogenic emissions fixed at present-day levels, the effects of climate change
alone on the concentrations of different PM$_{2.5}$ species display substantial seasonal
differences and spatial variability between the two periods. In winter, climate change alone is
found to decrease PM$_{2.5}$ concentration by as much as 12.0 μg m$^{-3}$ in the North China Plain,
but increase by up to 4.0 μg m$^{-3}$ in southeastern, northwestern and southwestern China.
These changes are mostly attributable to the changing chemistry and transport of different
species driven by changes in temperature, surface wind speed and planetary boundary layer
(PBL) depth. In summer, however, the changes of PM$_{2.5}$ display a generally opposite pattern
with increases (+6.0 to +8.0 μg m$^{-3}$) found in the North China Plain, and reductions (more
than -4.0 μg m$^{-3}$) found in most places of central and southern China, reflecting changes in
the same suite of meteorological variables but with varying relative importance. Climate
change alone leads to an increase in SOA concentration both in winter and summer (0.2-1.0

μg m$^{-3}$) in most of the eastern parts of China, primarily driven by enhanced biogenic VOC emissions resulting from warming.

The impacts of LCLU change alone on total PM$_{2.5}$ (-2.1 to +1.3 μg m$^{-3}$) is generally smaller than that of climate change alone, but the impacts on SOA and thus OC can be quite significant (-0.4 to +1.2 μg m$^{-3}$), reflecting the effects of deforestation, cropland expansion, reforestation as well as climate- and $CO_2$-driven changes in leaf area index (LAI). Changes in anthropogenic emissions from 1985 to 2005 levels are still the largest contributor to worsening PM$_{2.5}$ air quality in both summer and winter, leading to an increase in PM$_{2.5}$ by 54% on average in winter and 37% in summer over East Asia.

Our results show that the annual mean concentrations for PM$_{2.5}$ are in the range of 10-70 μg m$^{-3}$ in East Asia over 2007-2011 (Fig. S4). In many places of eastern and central China, and part of southwestern China, annual mean PM$_{2.5}$ well exceeds 35 μg m$^{-3}$, which is the limit value specified in the current air quality standards of the Ministry of Environmental Protection of China for all the cities nationwide except some special regions. The PM$_{2.5}$ air quality guideline set by the World Health Organization (WHO) is that annual mean concentration must not exceed 10 μg m$^{-3}$, which is even stricter (WHO, 2006). Beyond this level, the morbidity and premature mortality of health risks such as lung cancer, cardiovascular and respiratory diseases would increase in response to long-term exposure to PM$_{2.5}$. From this perspective, our results indicate that the effects of climate change would partly counteract the emission-driven increase in PM$_{2.5}$ in winter by a substantial fraction in most of northeastern, northern, eastern and central China especially in the North China Plain, imposing a so-called "climate benefit" for PM$_{2.5}$ air quality and thus public health. However, climate change could substantially exacerbate PM$_{2.5}$ pollution in summer in the North China Plain, northern and northwestern China, imposing a "climate penalty" instead. We also find that LCLU change might partially offset the increase in summertime PM$_{2.5}$ but further enhance wintertime PM$_{2.5}$ in the model through modifying the dry deposition of various PM$_{2.5}$ precursors and biogenic VOC emissions, which also act as important factors in modulating air quality.

There are various sources of uncertainties in this study. Previous work by Tai et al. (2013) suggested that the inclusion of $CO_2$ inhibition effect could reduce the sensitivity of surface SOA to climate and land cover changes in regions where isoprene emission is important, but this effect is not considered here. However, experimental data for $CO_2$-isoprene relationship at lower $CO_2$ levels are generally scarce and not robust enough to be included in our model periods. In addition, as pointed out by Fu and Tai (2015), vegetation

composition and resistance values for each vegetation or land type in this work are assumed to remain unchanged between 1980 and 2010, which may yield part of the uncertainties. The changes in manure and chemical fertilizer associated with the changes in agriculture practices and land use are also not taken into account in this study, which may affect soil $NO_x$ emission and contribute to the formation of inorganic $PM_{2.5}$, which may be particularly important in the future as anthropogenic $NO_x$ emissions are expected to decline. Furthermore, the deposition of $PM_{2.5}$ might also affect the terrestrial ecosystems and crops in various manners, e.g., via the acidification of soils that may lead to more leaching of mineral nutrients, and the introduction of excess nitrogen that may fertilize the soils or disrupt the soil nitrogen cycle. These processes would induce feedback effects that can further modify the land cover but are not explicitly taken into account in this study. All these issues remain poorly understood and warrant further investigation in future studies.

## Acknowledgements

This work was supported by the National Natural Science Foundation of China under grants 41405138 given to Yu Fu, and a start-up grant from the Croucher Foundation and The Chinese University of Hong Kong (6903601) given to Amos P. K. Tai. MERRA data used in this study/project have been have been provided by the Global Modeling and Assimilation Office (GMAO) at NASA Goddard Space Flight Center through the NASA GES DISC online archive. We acknowledge the MODIS LAI product provided by Land Processes Distributed Active Archive Center (LP DAAC). We also thank the Information Technology Services Centre (ITSC) at The Chinese University of Hong Kong for their devotion in providing the necessary computational services for this work.

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

**Table and Figure Captions**

Table 1. Summary of the simulations conducted in this study.

Figure 1. Seasonal mean surface concentrations of total $PM_{2.5}$, sulfate ($SO_4^{2-}$), nitrate ($NO_3^-$), ammonium ($NH_4^+$), organic carbon (OC), and black carbon (BC) in East Asia from the control ([*CTRL*]) simulation, averaged over 2007-2011.

Figure 2. Seasonal mean surface concentrations of total secondary organic aerosols (SOA), biogenic SOA, and anthropogenic SOA in East Asia from the control ([*CTRL*]) simulation, averaged over 2007-2011.

Figure 3. (a) Simulated vs. observed mean July-September SOA concentration. Observations are from Ding et al. (2014). Also shown is the 1:1 line (solid line) and linear fit (dashed), NMB is the normalized mean bias between simulated and observed concentrations; (b) Simulated vs. observed ratio of secondary organic carbon (SOC) to total organic carbon (OC) in China. The observed ratios are from Zhang et al. (2012). Also shown are the 1:1 line (solid line), 2:1 line and 1:2 line (dashed).

Figure 4. Simulated changes of wintertime (DJF) surface concentrations for $PM_{2.5}$, sulfate, nitrate, ammonium, organic aerosol, black carbon, surface temperature, total precipitation at the ground, relative humidity, surface wind speed, planetary boundary layer depth (PBL), and cloud fraction in East Asia arising from 1980-2010 changes in climate alone ([*CTRL*] − [*S_CLIM*]).

Figure 5. Simulated changes of summertime (JJA) surface concentrations for $PM_{2.5}$, sulfate, nitrate, ammonium, organic aerosol, black carbon, surface temperature, total precipitation at the ground, relative humidity, surface wind speed, planetary boundary layer depth (PBL), and cloud fraction in East Asia arising from 1980-2010 changes in climate alone ([*CTRL*] − [*S_CLIM*]).

Figure 6. Changes in surface secondary organic aerosol (SOA) concentration, isoprene emission, and monoterpene emission in winter (DJF) and summer (JJA) across East Asia arising from changes in climate alone ([*CTRL*] − [*S_CLIM*]) over 1980-2010.

Figure 7. Changes in seasonal mean surface concentrations of total $PM_{2.5}$. sulfate ($SO_4^{2-}$), nitrate ($NO_3^-$), ammonium ($NH_4^+$) and organic carbon (OC) in East Asia arising

from 1980-2010 changes in land cover and land use alone ([*CTRL*] – [*S_LCLU*]).

Figure 8. Changes in surface secondary organic aerosol (SOA) concentration, isoprene
emission, and monoterpene emission in winter (DJF) and summer (JJA) across
East Asia arising from 1980-2010 changes in land cover and land use alone
([*CTRL*] – [*S_LCLU*]).

Figure 9. Changes in seasonal (DJF and JJA) and annual (ANN) mean surface concentrations
of $PM_{2.5}$ and SOA in East Asia resulting from the combined effects of 1980-2010
changes in climate, land cover and land use ([*CTRL*] – [*S_COMB*]), and 1980-
2010 changes in anthropogenic emissions alone ([*CTRL*] - [*S_ANTH*]).


Table 1 Summary of the simulations conducted in this study.

| Simulations | MERRA Meteorology | Vegetation parameters | | Anthropogenic emissions |
| --- | --- | --- | --- | --- |
| | | Land cover * | Leaf area index | |
| CTRL | 2007-2011 | 2010 | 2010 | 2005 |
| S_CLIM | 1981-1985 | 2010 | 2010 | 2005 |
| S_LCLU | 2007-2011 | 1980 | 1982 | 2005 |
| S_COMB | 1981-1985 | 1980 | 1982 | 2005 |
| S_ANTH | 2007-2011 | 2010 | 2010 | 1985 |

* Land cover in terms of land or plant functional types.



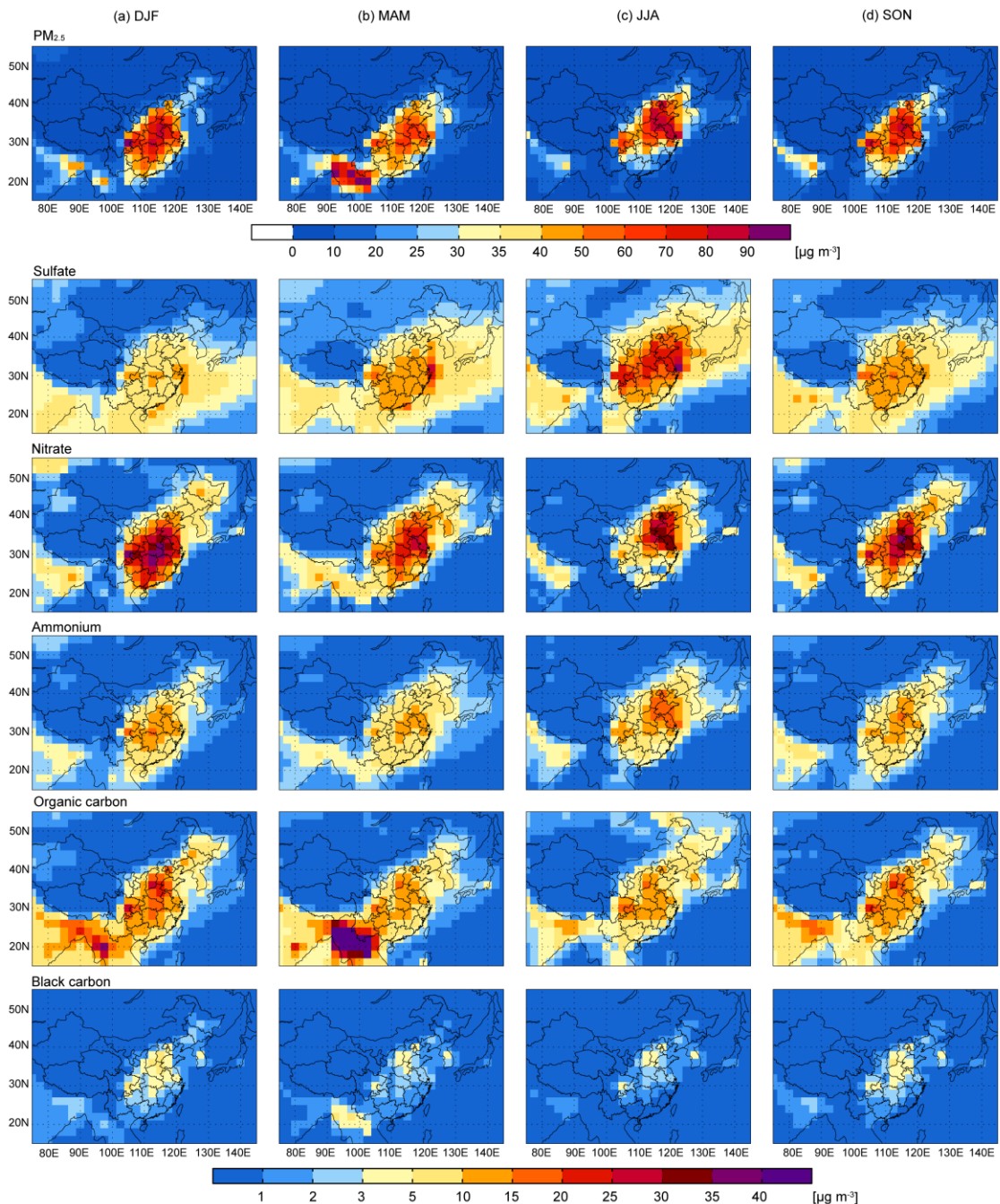


Fig. 1. Seasonal mean surface concentrations of total $PM_{2.5}$, sulfate ($SO_4^{2-}$), nitrate ($NO_3^-$), ammonium ($NH_4^+$), organic carbon (OC), and black carbon (BC) in East Asia from the control ([*CTRL*]) simulation, averaged over 2007-2011.



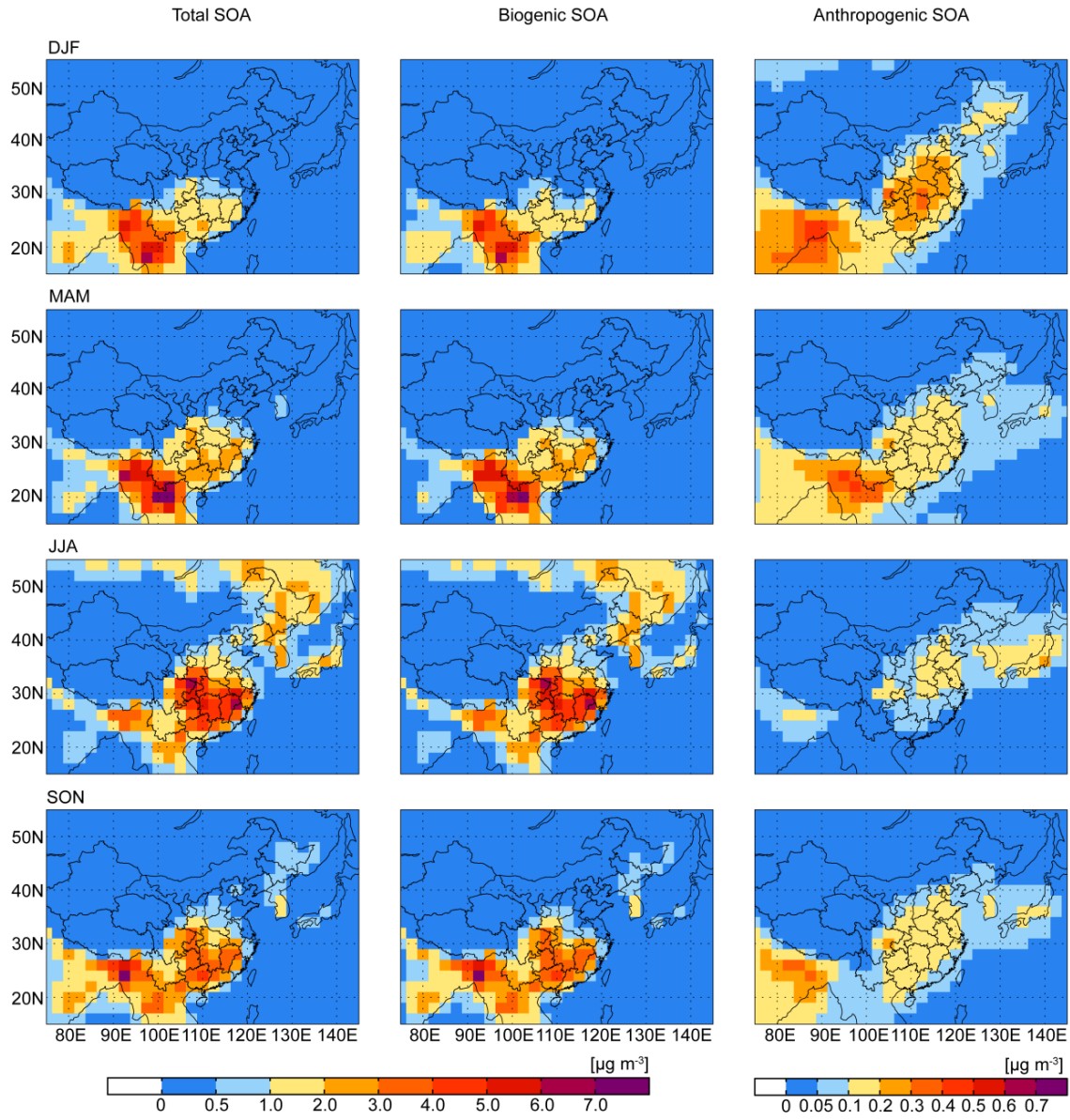


Fig. 2. Seasonal mean surface concentrations of total secondary organic aerosols (SOA),
biogenic SOA, and anthropogenic SOA in East Asia from the control ([*CTRL*]) simulation,
averaged over 2007-2011.


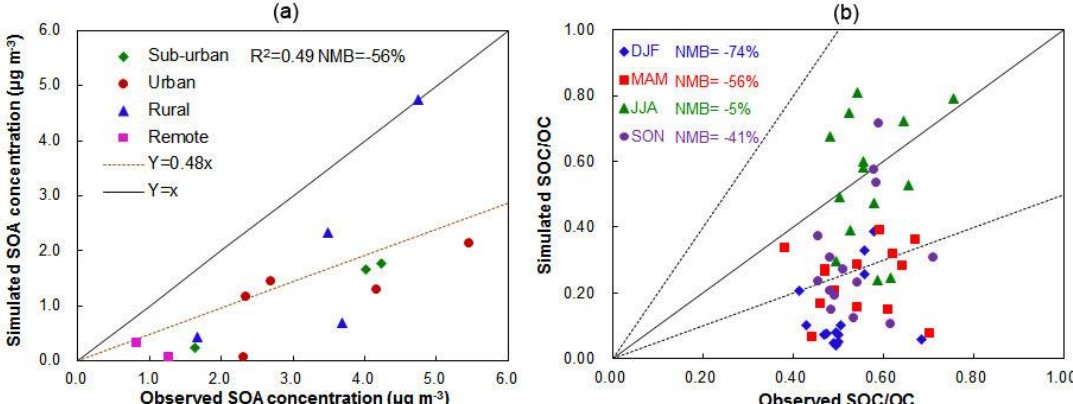


Fig. 3. (a) Simulated vs. observed mean July-September SOA concentration. Observations are from Ding et al. (2014). Also shown is the 1:1 line (solid line) and linear fit (dashed), NMB is the normalized mean bias between simulated and observed concentrations; (b) Simulated vs. observed ratio of secondary organic carbon (SOC) to total organic carbon (OC) in China. The observed ratios are from Zhang et al. (2012). Also shown are the 1:1 line (solid line), 2:1 line and 1:2 line (dashed).

708

709

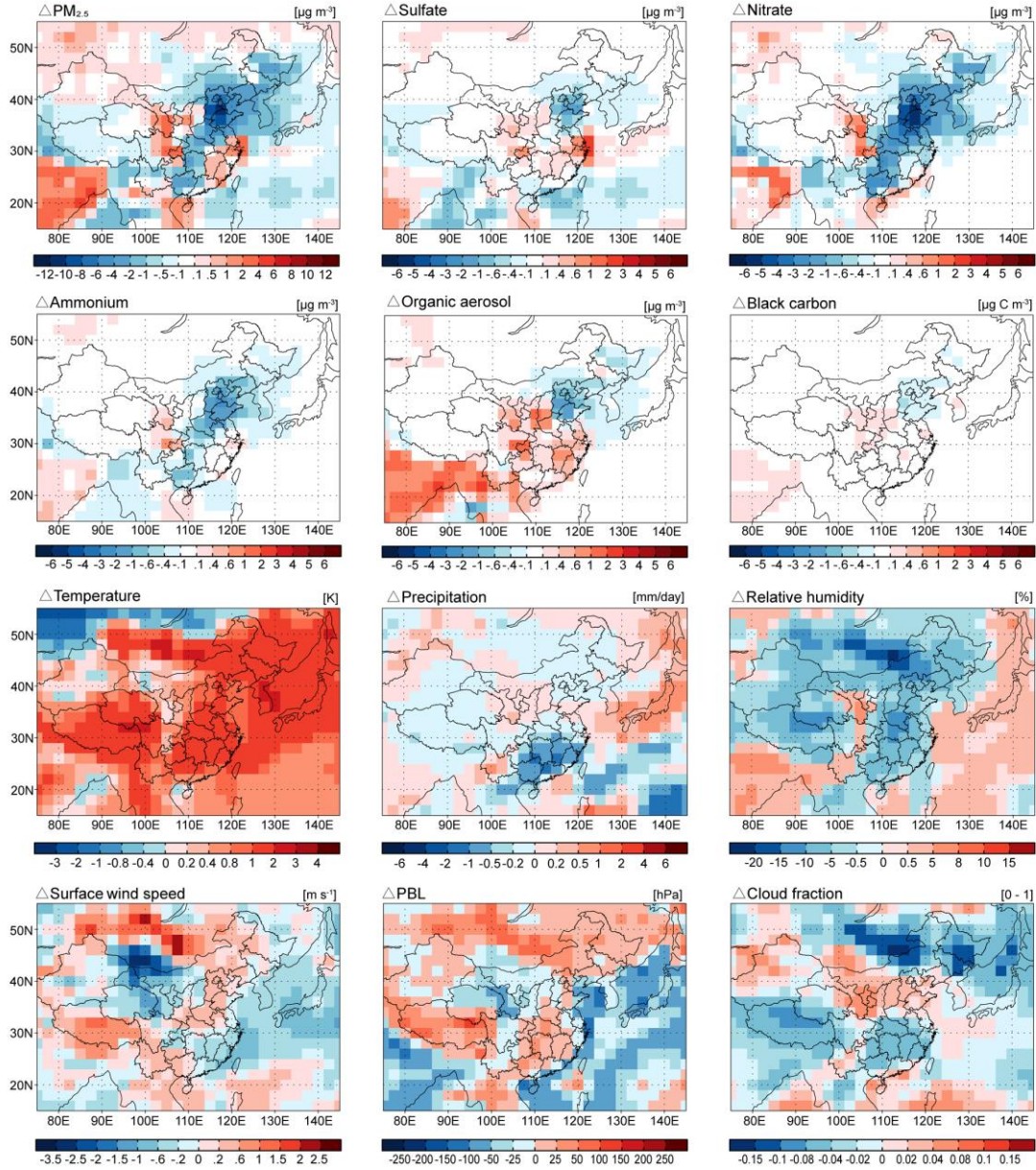

710

Fig. 4. Simulated changes of wintertime (DJF) surface concentrations for PM2.5, sulfate, nitrate, ammonium, organic aerosol, black carbon, surface temperature, total precipitation at the ground, relative humidity, surface wind speed, planetary boundary layer depth (PBL), and cloud fraction in East Asia arising from 1980-2010 changes in climate alone ([*CTRL*] − [*S_CLIM*]).



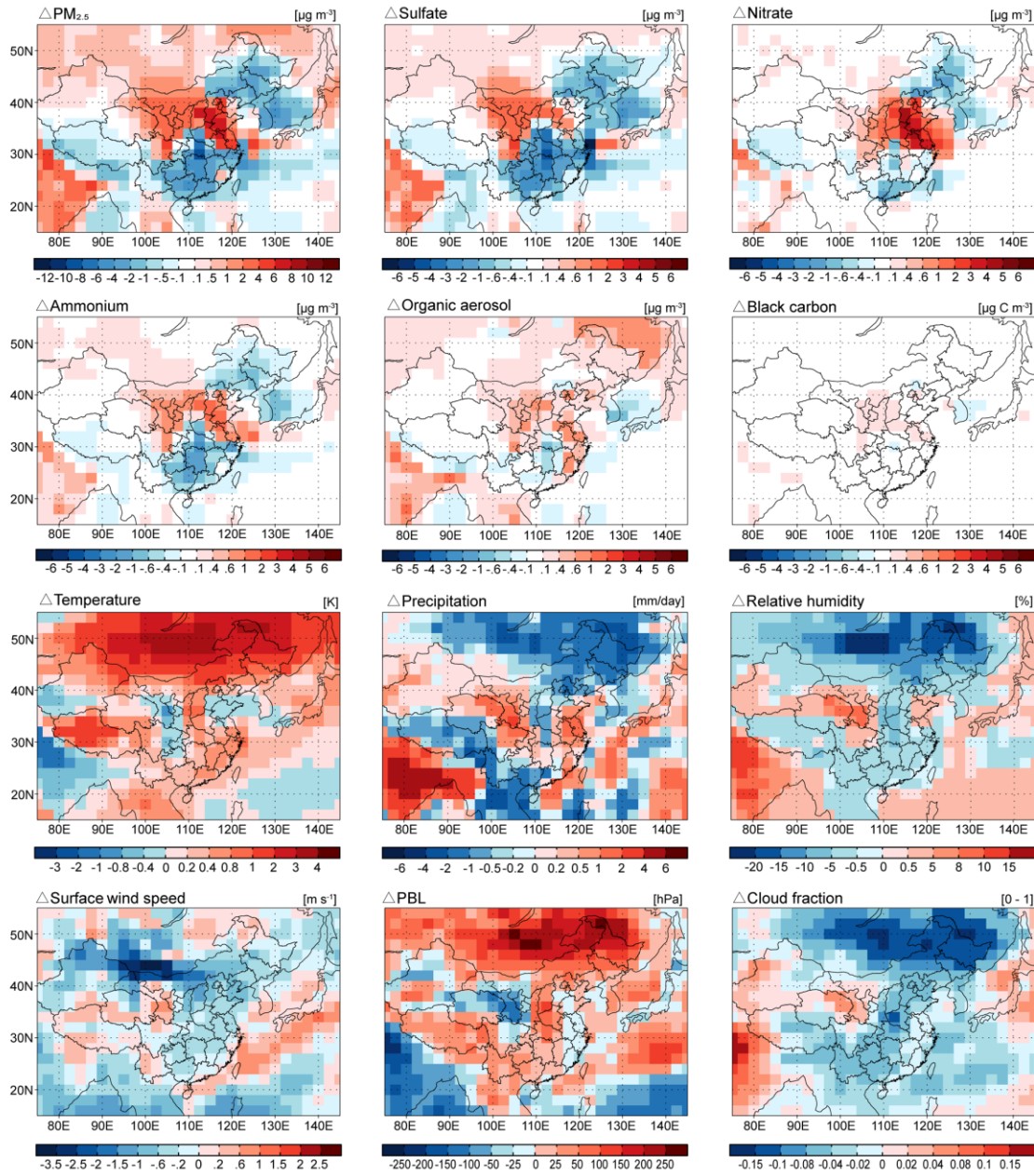

Fig. 5. Simulated changes of summertime (JJA) surface concentrations for PM$_{2.5}$, sulfate, nitrate, ammonium, organic aerosol, black carbon, surface temperature, total precipitation at the ground, relative humidity, surface wind speed, planetary boundary layer depth (PBL), and cloud fraction in East Asia arising from 1980-2010 changes in climate alone ([*CTRL*] − [*S_CLIM*]).


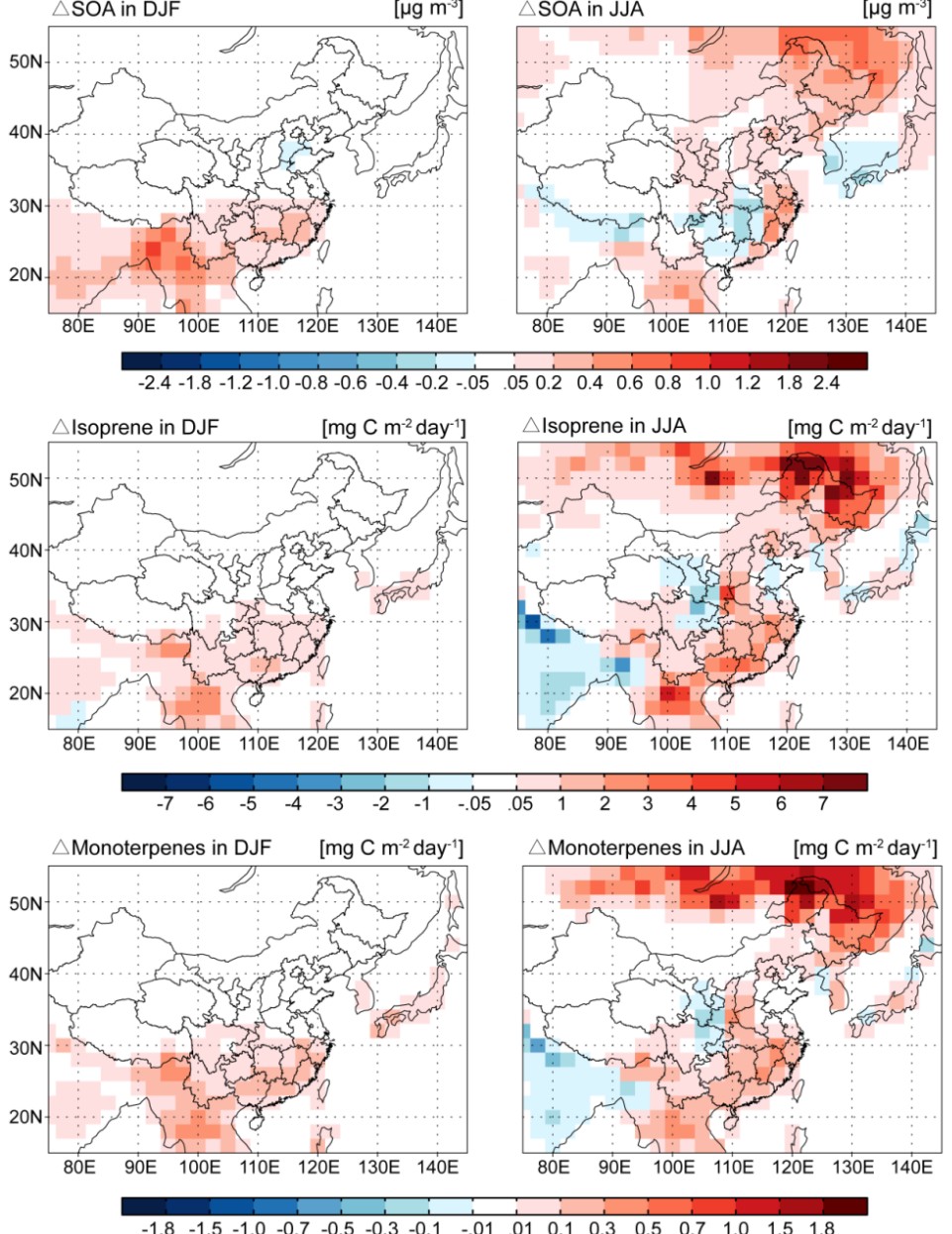

Fig. 6. Changes in surface secondary organic aerosol (SOA) concentration, isoprene
emission, and monoterpene emission in winter (DJF) and summer (JJA) across East Asia
arising from changes in climate alone ([*CTRL*] – [*S_CLIM*]) over 1980-2010.

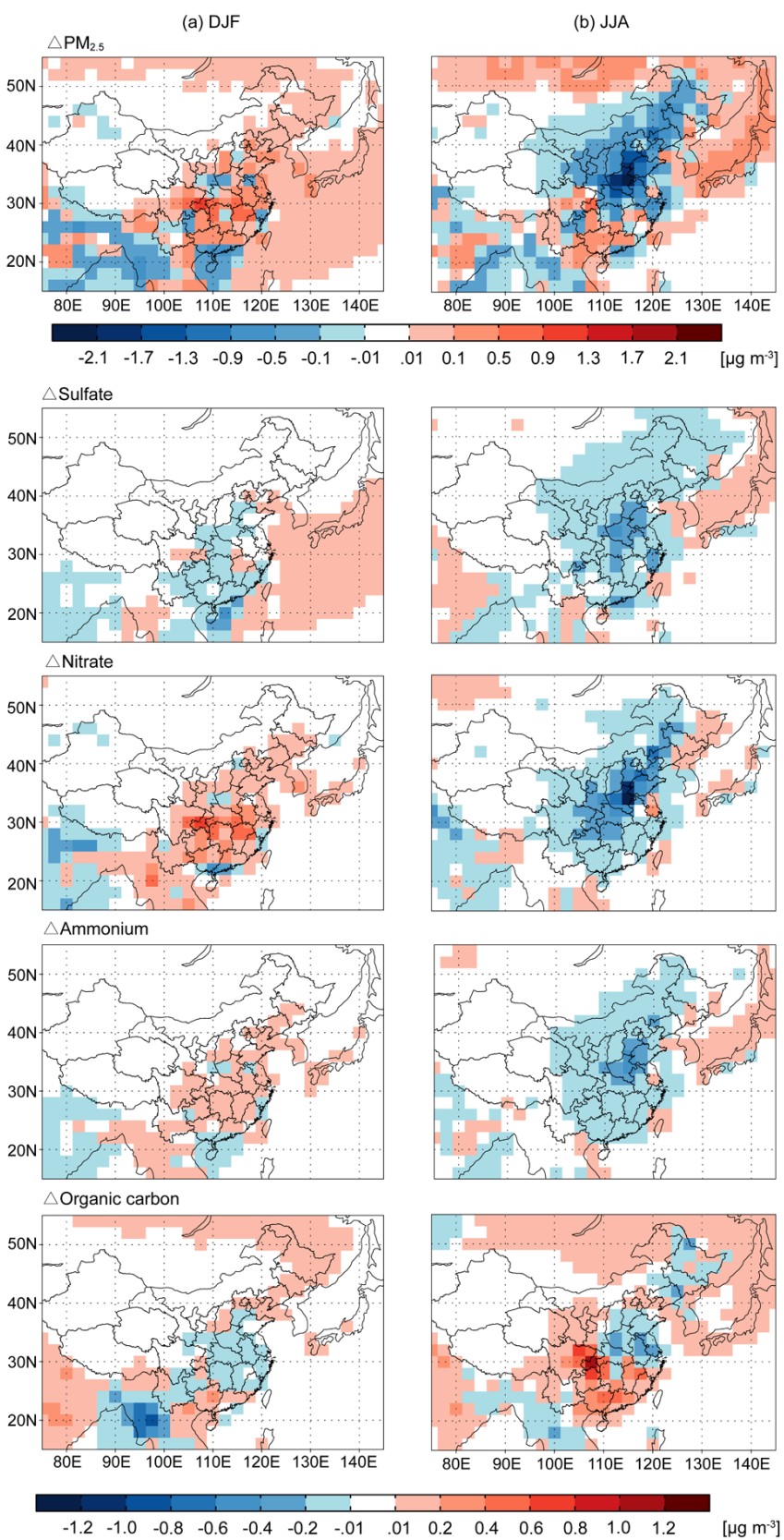

Fig. 7. Changes in seasonal mean surface concentrations of total PM$_{2.5}$. sulfate (SO$_4^{2-}$),
nitrate (NO$_3^-$), ammonium (NH$_4^+$) and organic carbon (OC) in East Asia arising from 1980-
2010 changes in land cover and land use alone ([*CTRL*] – [*S_LCLU*]).

734

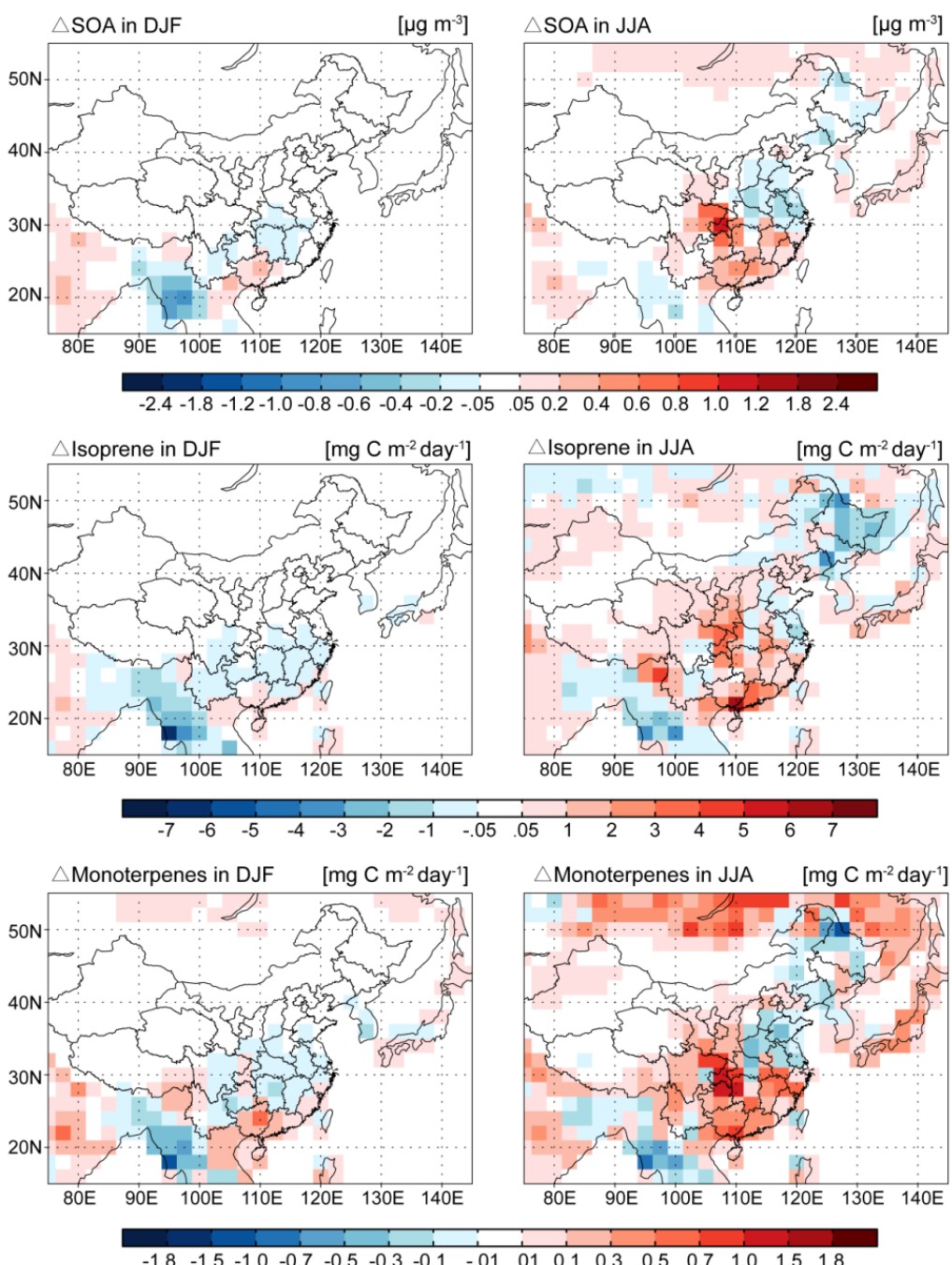

Fig. 8. Changes in surface secondary organic aerosol (SOA) concentration, isoprene emission, and monoterpene emission in winter (DJF) and summer (JJA) across East Asia arising from 1980-2010 changes in land cover and land use alone ([*CTRL*] − [*S_LCLU*]).

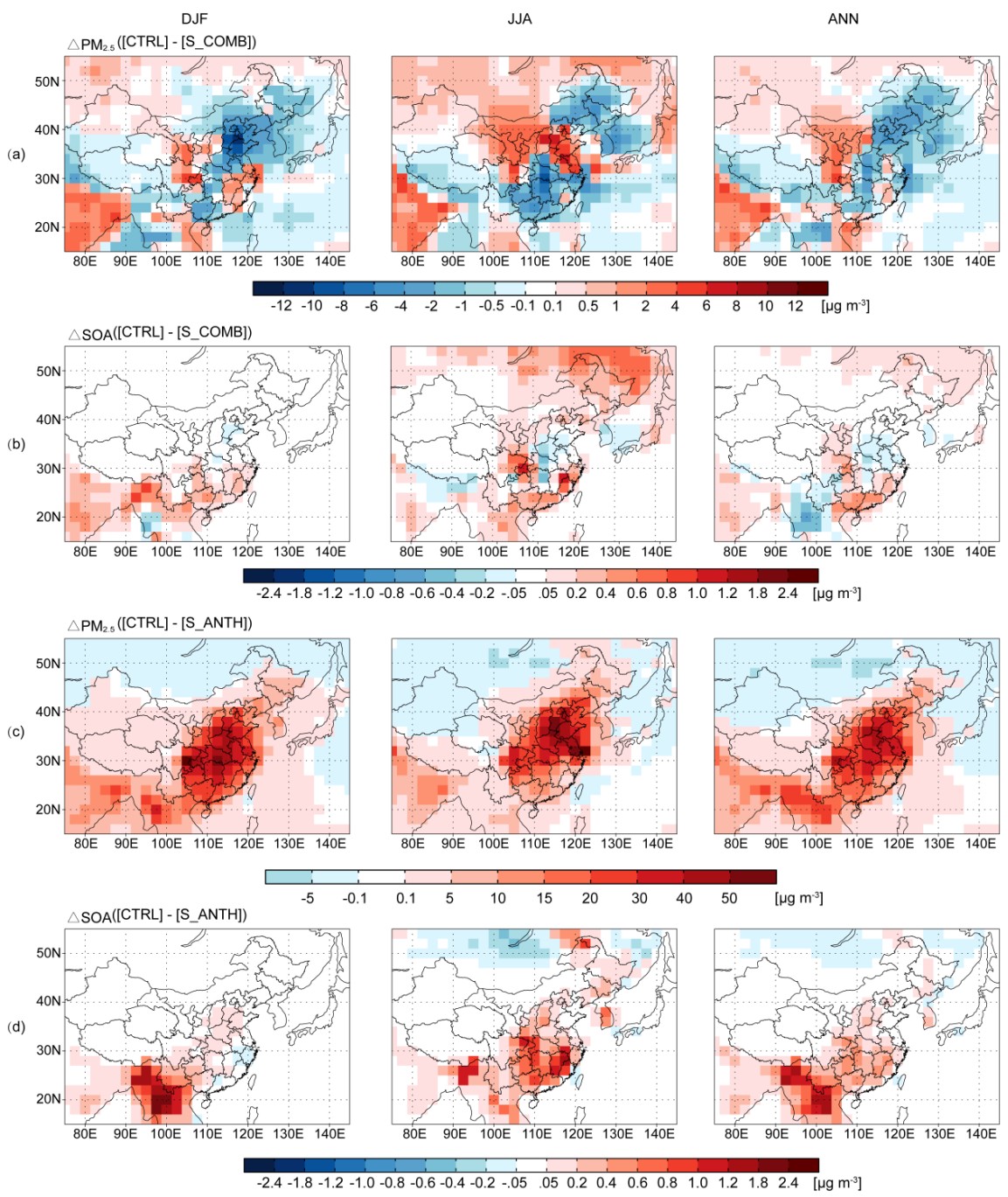

Fig. 9. Changes in seasonal (DJF and JJA) and annual (ANN) mean surface concentrations of PM$_{2.5}$ and SOA in East Asia resulting from the combined effects of 1980-2010 changes in climate, land cover and land use ([*CTRL*] – [*S_COMB*]), and 1980-2010 changes in anthropogenic emissions alone ([*CTRL*] – [*S_ANTH*]).