# Peer review of "Impacts of historical climate and land cover changes on fine particulate"

_Atmospheric Chemistry and Physics, 2016_

## Referee Comment (RC1) · Anonymous Referee #2 · 20 May 2016

This paper is virtually a follow up of previous works carried out by the same authors. The paper reports the effect of climate change, changes in land-use and land cover on PM2.5 and its precursors in east China. While this topic has been investigated quite extensively in the last several year, simultaneous influences of climate change and LULC on PM2.5 and corresponding chemical speciation were seldom tackled and provided useful information. The paper could be publishable in ACP. Before the paper goes to ACPD, some key fundamental issues should be addressed because these issues are crucial to the interpretation to their results.

Specific comments: 1. Authors selected 1981-1985 and 2007-2011 as their scenarios modeling periods. They should tell readers what was the rationale in the selection of

these two periods, and typical reasons? 2. To do so, authors need to present the differences of key meteorological / climate variables between the two periods, such as temperature, precipitation, and winds, e.g., T2010 - T1980 where T is gridded temperature over East Asia. They can present these differences like what they did in Fig. S1 of Supplementary. These would help readers to figure out where temperatures, wind speeds, and precipitation across East Asia increased or decreased for these two periods of time, and make sense to their discussions. 3. It might be better to present 5 model scenarios on a table with corresponding descriptions (line 170-180).

---

## Referee Comment (RC2) · Anonymous Referee #1 · 1 Jun 2016

Overview:

The study presented by Yu Fu and coworkers investigates the impact of changes in climate, land-use and anthropogenic emissions on fine particulate matter, using the GEOS-CHEM chemistry-transport model to carry out simulations for two 5-year periods (1981-1985 and 2007-2011).

The manuscript is concise, clearly presented and well written and of real interest, giving an interesting analysis of changes in PM2.5 over recent time slice. To clarify some of the aspects of the protocol and give a bit more perspective to this study, I would like the comments and corrections given below to be considered before the manuscript is published in ACP, which I strongly support.

[Figure]

General comments:

I think that the title is misleading as "over 1980-2010" suggests that the whole 1980-2010 period is investigated and presented, showing for instance also interannual variability, while two 5-year periods (1980s and 2010s) are actually considered. The title should therefore be modified accordingly.

The reason for choosing 1980s and 2010s for the study should be clearly demonstrated. What makes this period so interesting to investigate PM2.5 change in relation with land-use, climate or anthropogenic emission changes? Did strong changes in land-use occur in East Asia, in relation with management change? Were some important modifications of air quality regulations or industrial/human activities carried out? This would really help emphasizing the interest of choosing such a time line.

The close link with air quality is given in the title and clarified in the introduction but to me, the consequences of such changes in PM2.5 on air quality and health should be commented more in details, at list in the conclusion and discussion sections. How do PM2.5 levels for the different scenarios investigated compare with actual air quality thresholds (and what are the actual tolerance limits considered in East Asia)? This discussion could also be considered from both the human health and the vegetation point of view, as air pollution can be detrimental to both human beings and ecosystems.

The information regarding the fertilizer use, given in the conclusion and discussion section should also be given in the methods and model description. Please also detail which scenario was considered for fertilizer use, and the corresponding amount for the region of interest. Also, if a table giving the anthropogenic emissions for both scenarios, for a variety of key chemical species, is given in the supplementary material, no information is detailed regarding biogenic emissions of VOCs and NOx. And yet, those compounds are crucial considering the topic of the manuscript. Please also add a table, either in the main core of the manuscript of in the supplementary material, giving this information. When running the different simulations with GEOS-CHEM, at

which time-step where emission forcings, especially regarding biogenic emissions were considered? Monthly or higher? As biogenic VOC emissions are characterized by a strong diurnal variability, considering emissions at lower resolution could also impact the results regarding air quality and PM2.5 changes. This should be clarified and discussed.

Section 3, page 7, line 157: Please add some of the most important results from Zhang et al. (2012) for comparison.

Specific comments:

Abstract page 2, line 6: change "in northern China, but an increase in summertime" by "in northern China, but to an increase in summertime"

Page 3, line 11: change "would be useful to help better project" to "would be useful to help to better project"

Page 3, line 15: change "This attribution of East Asian air quality" to "This attribution of East Asia air quality"

Page 5, line 88: please write CTM in full, as used for the first time in the text Page 6, line 116: change "inputs, namely, leaf area index (LAI), and land" to "inputs, namely leaf area index (LAI) and land" removing commas

Page 7, line 157: change "Zhang et al " to Zhang et al. "

Page 9, line 207: change Âń uncertainties in not only Âż to Âń uncertainties not only in Âż

Page 9, line 234: change "but depending on region changes in wind speed" to "but depending on the region, changes in wind speed"

Page 10, line 236: something is missing in the sentence. "might have substantially enhanced or partly counteracted"... enhanced or counteracted what exactly?

Page 10, line 237: change "the same patterns as that for sulfate" to "the same patterns as those for sulfate"

Page 10, line 245: change "Increased temperature also in part contributes" to "Increased temperature also partly contributes"

---

## Author Comment (AC1) · 25 Jul 2016

**Responses to Reviewers on " Impacts of historical climate and land cover changes on fine particulate matter (PM$_{2.5}$) air quality in East Asia over 1980-2010" by Y. Fu, A. P. K. Tai, and H. Liao. (MS No.: acp-2016-299)**

We would like to thank the reviewer for the thoughtful comments. The manuscript has been revised accordingly, and our point-by-point responses are provided below. The reviewers' comments are *italicized*, and our new/modified text is highlighted in **bold** below, and highlighted in blue in the manuscript.

**Response to Anonymous Referee #1**

*The manuscript is concise, clearly presented and well written and of real interest, giving an interesting analysis of changes in PM$_{2.5}$ over recent time slice. To clarify some of the aspects of the protocol and give a bit more perspective to this study, I would like the comments and corrections given below to be considered before the manuscript is published in ACP, which I strongly support.*

*General comments:*

*I think that the title is misleading as "over 1980-2010" suggests that the whole 1980-2010 period is investigated and presented, showing for instance also interannual variability, while two 5-year periods (1980s and 2010s) are actually considered. The title should therefore be modified accordingly.*

As the reviewer's suggested, the title is changed to "**Impacts of historical climate and land cover changes on fine particulate matter (PM$_{2.5}$) air quality in East Asia between 1980 and 2010**".

*The reason for choosing 1980s and 2010s for the study should be clearly demonstrated. What makes this period so interesting to investigate PM$_{2.5}$ change in relation with land-use, climate or anthropogenic emission changes? Did strong changes in land-use occur in East Asia, in relation with management change? Were some important modifications of air quality regulations or industrial/human activities carried out? This would really help emphasizing the interest of choosing such a time line.*

The simulations in our study account for the effects of changes in meteorological variables, the land cover and land use from early 1980s to late 2000s according to a single and coherent set of assimilated meteorology data (MERRA), which covers the period 1979-present, so the periods we chose represent approximately the longest time lapse possible to examine.

We extend the description in relation to the reasons for studying these two periods in Sect. 1 (Page 4, L121-L131): "**Because of climate change, rapid economic development and other human activities, LCLU in East Asia has undergone remarkable changes in the past 30 years. Especially in China, some major economic reforms and land use policies have been implemented since December 1978, which together with simultaneous changes in climate have resulted in a whirlwind of changes in the terrestrial environments of China. Based on satellite-derived images and surveys, LULC changes in China from late 1980s to the mid-2000s are characterized by an expansion of urban areas, deserts and bare lands, and an overall decrease in vegetation coverage (Fu and Liao, 2014).** In this study, we use the GEOS-Chem global **chemical transport model (CTM)** driven by **past land cover data and meteorological fields from a**

**single and coherent set of assimilated meteorology (MERRA), which covers the period 1979-present. We** quantify the impacts of historical changes in climate, land cover and land use on $PM_{2.5}$ air quality in East Asia between two 5-year periods: historical period, 1981-1985 (referred to as "1980"), and the present day, 2007-2011 (referred to as "2010")."

*The close link with air quality is given in the title and clarified in the introduction but to me, the consequences of such changes in $PM_{2.5}$ on air quality and health should be commented more in details, at list in the conclusion and discussion sections. How do $PM_{2.5}$ levels for the different scenarios investigated compare with actual air quality thresholds (and what are the actual tolerance limits considered in East Asia)? This discussion could also be considered from both the human health and the vegetation point of view, as air pollution can be detrimental to both human beings and ecosystems.*

Current air quality standards published by the environment ministry of China, Japan, and South Korea specify that for all cities nationwide except some special regions such as national parks, the limit values of annual mean $PM_{2.5}$ are 35 μg m$^{-3}$, 15 μg m$^{-3}$, 25 μg m$^{-3}$, respectively. Based on known health effects, the World Health Organization (WHO) has set a guideline value for $PM_{2.5}$, which states that annual average concentration must not exceed 10 μg m$^{-3}$. This guideline value represents the lowest level beyond which the morbidity and premature mortality of health diseases such as lung cancer, cardiovascular and respiratory diseases have been shown to increase in response to long-term exposure to $PM_{2.5}$ (WHO, 2006).

We have added some discussion in Sect.7. (Page 13, L410-L422) "**Our results show that the annual mean concentrations for $PM_{2.5}$ are in the range of 10-70 μg m$^{-3}$ in East Asia over 2007-2011 (Fig. S4). In many places of eastern and central China, and part of southwestern China, annual mean $PM_{2.5}$ well exceeds 35 μg m$^{-3}$, which is the limit value specified in the current air quality standards of the Ministry of Environmental Protection of China for all the cities nationwide except some special regions. The $PM_{2.5}$ air quality guideline set by the World Health Organization (WHO) is that annual mean concentration must not exceed 10 μg m$^{-3}$, which is even stricter (WHO, 2006). Beyond this level, the morbidity and premature mortality of health risks such as lung cancer, cardiovascular and respiratory diseases would increase in response to long-term exposure to $PM_{2.5}$. From this perspective, our results indicate that the effects of climate change would partly counteract the emission-driven increase in $PM_{2.5}$ in winter by a substantial fraction in most of northeastern, northern, eastern and central China especially in the North China Plain, imposing a so-called "climate benefit" for $PM_{2.5}$ air quality and thus public health. ……**"

A discussion on $PM_{2.5}$-vegetation interactions is also included in the last paragraph.

(Sect.7, Page 14, L441-L445): "**Furthermore, the deposition of $PM_{2.5}$ might also affect the terrestrial ecosystems and crops in various manners, e.g., via the acidification of soils that may lead to more leaching of mineral nutrients, and the introduction of excess nitrogen that may fertilize the soils or disrupt the soil nitrogen cycle. These processes would induce feedback effects that can further modify the land cover but are not explicitly taken into account in this study. All t**hese issues remain poorly understood and warrant further investigation in future studies."

[Figure]

Figure S4. Annual mean concentration for PM$_{2.5}$ from simulations [*CTRL*], [*S_COMB*], and [*S_ANTH*].

*The information regarding the fertilizer use, given in the conclusion and discussion section should also be given in the methods and model description. Please also detail which scenario was considered for fertilizer use, and the corresponding amount for the region of interest.*

We now include this in the main text as follow: (Sect. 2, Page 6, L169-172): "**The reservoir of nitrogen associated with manure and chemical fertilizer remains unchanged between 1980 and 2010 by using the fixed inventory for fertilizer and manure emissions from Potter et al. (2010).**"
The amount of NO$_x$ emission from fertilizer use are shown in the supplementary materials (Table S3, see below).

*Also, if a table giving the anthropogenic emissions for both scenarios, for a variety of key chemical species, is given in the supplementary material, no information is detailed regarding biogenic emissions of VOCs and NO$_x$. And yet, those compounds are crucial considering the topic of the manuscript. Please also add a table, either in the main core of the manuscript of in the supplementary material, giving this information.*

Table S3 in the supplementary material now includes the biogenic VOCs emissions and soil NO$_x$ emission.
**Table S3. Biogenic hydrocarbon emission, and soil and fertilizer NO$_x$ emission in [CTRL], [S_LCLU], [S_CLIM], and [S_COMB] simulations in East Asia (15 °-55 °N, 80 °-145 °E).**

| Species | CTRL | S_LCLU (changes, %) | S_CLIM (changes, %) | S_COMB (changes, %) |
|---|---|---|---|---|
| **Biogenic hydrocarbons (Tg C yr$^{-1}$)** | | | | |
| Isoprene | 25.28 | 26.28 (-3.8) | 23.09 (+9.5) | 24.15 (+4.7) |
| Monoterpenes | 6.98 | 7.03 (-0.7) | 6.53 (+6.9) | 6.61 (+5.6) |
| Others VOC[*] | 5.69 | 5.78 (-1.6) | 5.31 (+7.2) | 5.42 (+5.0) |
| Total | 37.94 | 39.10 (-3.0) | 34.93 (+8.6) | 36.18 (+4.9) |
| **NO$_x$ (Tg N yr$^{-1}$)** | | | | |
| Soil NO$_x$ | 1.32 | 1.31 (+0.3) | 1.19 (+10.3) | 1.19 (+10.7) |
| Fertilizer NO$_x$ | 0.51 | 0.51 (+0.1) | 0.49 (+4.0) | 0.49 (+4.1) |

[*] Others VOCs include methyl butenol, farnesene, b-caryophyllene, other sesquiterpenes, other monoterpenes, acetone, and lumped ⩾ C3 alkenes.

*When running the different simulations with GEOS-CHEM, at which time-step where emission forcings, especially regarding biogenic emissions were considered? Monthly or higher? As biogenic VOC emissions are characterized by a strong diurnal variability, considering emissions at lower resolution could also impact the results regarding air quality and PM$_{2.5}$ changes. This should be clarified and discussed.*

As pointed by the reviewer, biogenic VOC emissions, which are the major precursors in forming secondary organic aerosol, are characterized by a strong diurnal variation. In this work, the emissions time-step is set to 30 minutes in all the GEOS-Chem simulations, and the biogenic VOC emissions are calculated online by the MEGAN module (the model of Emissions of Gasses and Aerosol from Nature), which is embedded in GEOS-Chem. Therefore, the diurnal variability of biogenic VOC emissions is included in all the simulation.

To clarify, we now include a brief description in the main text as follows (Page 5, L164-L166): "The emissions of biogenic VOC species in each grid cell ……, using the **online** Model of Emissions of Gases and Aerosols from Nature (MEGAN) module **(Guenther et al., 2012) embedded in GEOS-Chem with the emissions time-step set to 30 minutes**."

*Section 3, page 7, line 157: Please add some of the most important results from Zhang et al. (2012) for comparison.*

We now include these information in Sect. 3 (Page 7, L201-L204): "**Based on the measurements from 16 sites in China, Zhang et al. (2012) reported that sulfate (~16%), OC (~15%), nitrate (~7%), ammonium (~5%) and mineral aerosol (~35%) are majorities of the total PM$_{10}$ concentration.**"

*Specific comments:*

*Abstract page 2, line 6: change "in northern China, but an increase in summertime" by "in northern China, but to an increase in summertime"*

(Page 1, L16) Changed.

*Page 3, line 11: change "would be useful to help better project" to "would be useful to help to better project"*

(Page 2, L44) Grammatically speaking "help" can be followed by both bare infinitive (without "to") and to-infinitive (with "to"), so the original form we wrote was not wrong. But this is changed anyway.

*Page 3, line 15: change "This attribution of East Asian air quality" to "This attribution of East Asia air quality"*

(Page 2, L48) Changed.

*Page 5, line 88: please write CTM in full, as used for the first time in the text*

(Page 3, L99) We have changed "CTM" to "chemical transport model (CTM)"

*Page 6, line 116: change "inputs, namely, leaf area index (LAI), and land" to "inputs, namely leaf area index (LAI) and land" removing commas*

(Page 5, L159) Revised.

*Page 7, line 157: change "Zhang et al " to Zhang et al. "*

(Page 7, L205) Changed.

*Page 9, line 207: change " uncertainties in not only" to " uncertainties not only in "*

(Page 8, L251) Changed.

*Page 9, line 234: change "but depending on region changes in wind speed" to "but depending on the region, changes in wind speed"*

(Page 9, L283) Changed.

*Page 10, line 236: something is missing in the sentence. "might have substantially enhanced or partly counteracted": : : enhanced or counteracted what exactly?*

(Page 9, L285-L286) The sentence is changed to "might have either substantially enhanced (most of northeastern, northern and central China) or partly counteracted (e.g., southeastern China) **the reduction in ammonium nitrate,** ……".

*Page 10, line 237: change "the same patterns as that for sulfate" to "the same patterns as those for sulfate"*

(Page 9, L287) Changed.

*Page 10, line 245: change "Increased temperature also in part contributes" to "Increased temperature also partly contributes"*

(Page 9, L295) Changed.

**References**

Potter, P., Ramankutty, N., Bennett, E. M., and Donner, S. D.: Characterizing the Spatial Patterns of Global Fertilizer Application and Manure Production, Earth Interact., 14, 1-22, doi:10.1175/2009EI288.1, 2010.

World Health Organization (WHO): WHO air quality guidelines for particulate matter, ozone, nitrogen dioxide and sulfur dioxide-Global update 2005-Summary of risk assessment. WHO, Geneva, Switzerland, 9-13 pp., 2006.

**Response to Anonymous Referee #2**

*This paper is virtually a follow up of previous works carried out by the same authors. The paper reports the effect of climate change, changes in land-use and land cover on PM₂.₅ and its precursors in east China. While this topic has been investigated quite extensively in the last several year, simultaneous influences of climate change and LULC on PM₂.₅ and corresponding chemical speciation were seldom tackled and provided useful information. The paper could be publishable in ACP. Before the paper goes to ACPD, some key fundamental issues should be addressed because these issues are crucial to the interpretation to their results.*

*Specific comments:*

*1. Authors selected 1981-1985 and 2007-2011 as their scenarios modeling periods. They should tell readers what was the rationale in the selection of these two periods, and typical reasons?*

> We now include this information in the main text in Sect. 1 (Page 4, L121-L131): "**Because of climate change, rapid economic development and other human activities, LCLU in East Asia has undergone remarkable changes in the past 30 years. Especially in China, some major economic reforms and land use policies have been implemented since December 1978, which together with simultaneous changes in climate have resulted in a whirlwind of changes in the terrestrial environments of China**. **Based on satellite-derived images and surveys, LULC changes in China from late 1980s to the mid-2000s are characterized by an expansion of urban areas, deserts and bare lands, and an overall decrease in vegetation coverage (Fu and Liao, 2014). In this study,** we use the GEOS-Chem global **chemical transport model (CTM)** driven by **past land cover data and meteorological fields from a single and coherent set of assimilated meteorology (MERRA), which covers the period 1979-present. We** quantify the impacts of historical changes in climate, land cover and land use on PM₂.₅ air quality in East Asia between two 5-year periods: historical period, 1981-1985 (referred to as "1980"), and the present day, 2007-2011 (referred to as "2010")."

*2. To do so, authors need to present the differences of key meteorological / climate variables between the two periods, such as temperature, precipitation, and winds, e.g., T2010 - T1980 where T is gridded temperature over East Asia. They can present these differences like what they did in Fig. S1 of Supplementary. These would help readers to figure out where temperatures, wind speeds, and precipitation across East Asia increased or decreased for these two periods of time, and make sense to their discussions.*

> (Page 27-28) The differences of key meteorological variables between the two periods including temperature (△temperature), relative humidity (△relative humidity), precipitation (△precipitation), wind speed (△surface wind speed), mixing height (△PBL) and cloud fraction (△cloud fraction) are included in our manuscript (Fig. 4 and Fig. 5, for winter and summer, respectively).

*3. It might be better to present 5 model scenarios on a table with corresponding descriptions (line 170-180).*

The table suggested is now added as Table 1 in the main text (Page 23).

**Table 1. Summary of the simulations conducted in this study.**

| Simulations | MERRA meteorology | Vegetation parameters | | Anthropogenic emissions |
|---|---|---|---|---|
| | | Land cover [*] | Leaf area index | |
| CTRL | 2007-2011 | 2010 | 2010 | 2005 |
| S_CLIM | 1981-1985 | 2010 | 2010 | 2005 |
| S_LCLU | 2007-2011 | 1980 | 1982 | 2005 |
| S_COMB | 1981-1985 | 1980 | 1982 | 2005 |
| S_ANTH | 2007-2011 | 2010 | 2010 | 1985 |

[*] Land cover in terms of land or plant functional types.